# Configurable Readout Error Mitigation in Quantum Workflows

**Martin Beisel ***[ID]**, Johanna Barzen** [ID]**, Frank Leymann** [ID]**, Felix Truger** [ID]**, Benjamin Weder** [ID]
**and Vladimir Yussupov** [ID]

Institute of Architecture of Application Systems, University of Stuttgart, Universitätsstraße 38,
70569 Stuttgart, Germany
* Correspondence: martin.beisel@iaas.uni-stuttgart.de

**Abstract:** Current quantum computers are still error-prone, with measurement errors being one of the factors limiting the scalability of quantum devices. To reduce their impact, a variety of readout error mitigation methods, mostly relying on classical post-processing, have been developed. However, the application of these methods is complicated by their heterogeneity and a lack of information regarding their functionality, configuration, and integration. To facilitate their use, we provide an overview of existing methods, and evaluate general and method-specific configuration options. Quantum applications comprise many classical pre- and post-processing tasks, including readout error mitigation. Automation can facilitate the execution of these often complex tasks, as their manual execution is time-consuming and error-prone. Workflow technology is a promising candidate for the orchestration of heterogeneous tasks, offering advantages such as reliability, robustness, and monitoring capabilities. In this paper, we present an approach to abstractly model quantum workflows comprising configurable readout error mitigation tasks. Based on the method configuration, these workflows can then be automatically refined into executable workflow models. To validate the feasibility of our approach, we provide a prototypical implementation and demonstrate it in a case study from the quantum humanities domain.

**Keywords:** quantum computing; survey; readout error mitigation; measurement errors; workflows

## 1. Introduction

Quantum computing promises breakthroughs in many fields, e.g., machine learning, chemistry, and optimization [1–3]. Taking advantage of quantum mechanical phenomena such as superposition and entanglement, quantum algorithms may outperform their classical counterparts regarding speed, accuracy, or energy efficiency [3,4]. Although quantum devices have improved significantly over the past few years, they are still noisy, error-prone, and provide a limited number of qubits [5]. The errors caused by different sources, e.g., error-prone gate executions or faulty measurements, lead to inaccurate results [6]. However, contrary to classical bits, the state of a qubit can not be copied [7], making classical error handling methods infeasible for quantum computing. Thus, new error handling mechanisms are being introduced for *correcting* or *mitigating* quantum errors [8–11].

Error correction enables fault-tolerant quantum computation by performing an in-flight repair of computational errors [8]. However, correcting quantum errors requires a large number of qubits and significantly increases circuit depth, making it infeasible for the current generation of *Noisy Intermediate-Scale Quantum (NISQ)* devices [3]. In contrast, error mitigation methods focus on mitigating the impact of errors with little to no modifications in quantum circuits, i.e., the circuit depth and width remain similar. For example, *Gate Error Mitigation (GEM)* methods [12–14] adapt quantum circuits before their execution to compensate for expected gate errors, and *Readout Error Mitigation (REM)* methods [15–17] often rely on classical post-processing to reduce the impact of measurement errors [8].

Due to the high measurement error rates of many NISQ devices [18], REM becomes a necessity for executing quantum circuits [15]. While a variety of methods for mitigating

measurement errors have been proposed in the literature [19–21], their configuration differs significantly, e.g., the specification of method-specific attributes, and requires specific expertise in different domains, in addition to having a knowledge of quantum computers and error mitigation methods, which can hinder their application in practice. For example, some REMs rely on training neural networks [22], whereas others employ arithmetic matrix operations [15]. Hence, they impose different configuration and integration requirements on quantum application developers. Furthermore, acquiring a list of existing methods and their categorization to facilitate selecting "the best tool for the job" is difficult and time-consuming, as most methods are introduced in highly detailed papers. Therefore, quantum application developers must often choose from a wide range of methods with different advantages, disadvantages, and requirements. This requires an understanding of (i) what makes existing methods different and (ii) how to configure them correctly prior to integrating them into quantum applications. This leads us to Challenge 1: *"The lack of centralized documentation for REM methods makes it non-trivial for quantum software engineers to explore, adopt, and exchange the method implementations. Furthermore, existing works do not use a uniform terminology and structure, making it difficult to compare different methods. This particularly applies to heterogeneous configuration options of existing methods, further complicating the integration of REM into quantum applications".*

Moreover, quantum circuits are typically not executed independently, but rather as part of a complex process that comprises many tasks [23,24]. Many of these tasks are performed on classical hardware, e.g., database access, user interaction, or data preparation for the quantum circuit, while others are executed on quantum hardware. Hence, their heterogeneity leads to several prerequisites, e.g., different programming languages, operating systems, or hardware. Therefore, the execution of quantum applications is a hybrid process consisting of various quantum and classical tasks [23]. The manual integration and execution of these tasks is time-consuming and error-prone as it requires knowledge from different domains, e.g., quantum software engineering, integration, and deployment automation. One proven way to enable such an integration is to use *workflow technology*, which enables the robust and scalable orchestration of complex compositions of heterogeneous tasks [25,26]. As a step towards workflow-based modeling and the execution of quantum applications, Weder et al. [27] introduced a quantum-specific modeling extension for workflow languages. For example, it can be used to graphically model and execute typical quantum tasks, such as circuit execution or REM, in *Business Process Model and Notation (BPMN)* [28]. However, to the best of our knowledge, there exist no works focusing on the issue of automating the configuration of REM methods, as well as in the context of workflow-based modeling and the execution of quantum applications, which leads us to Challenge 2: *"The manual integration of a hybrid quantum application into existing software systems is complex, time-consuming, inefficient, and error-prone. Particularly reoccurring tasks with changing configurations such as REM can benefit from automation. While there are concepts that enable modeling quantum applications using workflows, there exist no concepts and techniques for automating the configuration and execution of REM".*

In this work, we address these challenges by (i) conducting a survey of existing REM methods to categorize them and to identify the underlying method-specific and method-agnostic configuration options. This analysis supports quantum application developers in deciding on suitable methods, and facilitates the configuration of the chosen method when specifying the workflow model. Therefore, to automate the configuration of REM methods, we (ii) present a model-driven configuration approach that enables the automation of the execution of REM in quantum workflows. To validate the feasibility of our approach, we (iii) implement our concepts prototypically by extending an existing open-source framework for modeling, transforming, and deploying quantum workflows [29]. Finally, we (iv) conduct a case study from the quantum humanities domain showcasing the automated configuration of the REM process using our prototypical implementation.

The remainder of this paper is organized as follows: Section 2 discusses the fundamental concepts and Section 3 presents the survey of REM methods. Section 4 introduces

our approach for the automated configuration of REM in quantum workflows. Section 5 elaborates on the case study from the quantum humanities domain. Finally, Sections 6 and 7 discuss our findings and related work, and Section 8 concludes the paper.

## 2. Background

In this section, we discuss the relevant background on quantum error handling and on the workflow-based execution of hybrid quantum applications.

### 2.1. Quantum Errors and Their Handling in the NISQ Era

Current NISQ devices are error-prone, reducing the result quality and limiting the scalability of quantum computations [3]. The reason for the high number of errors is the fragility and complexity of quantum systems which leads to unintended quantum state changes. Moreover, quantum errors originate from various sources, e.g., qubits themselves are unstable and decay over short periods of time; so-called *decoherence*. Furthermore, unintentional state changes happen due to erroneous gate executions [12,30–32], or the unavoidable coupling between qubits and their physical environment [33].

Although error-prone qubits and gate errors play a significant role, *measurement errors* often play a crucial role too [15]. Measurement errors, also called readout errors, are caused by erroneous measurement operations and the significant measurement times, which can lead to the decoherence of quantum states during the measurement [34]. As a consequence, the measurement result deviates from the prepared quantum state [5]. To reduce the impact of measurement errors, several REM methods have been developed [15,16,19]. Figure 1 shows a typical process when applying REM. During quantum circuit execution, the solution state is prepared and finally measured to obtain its information. However, measuring a qubit's state collapses the state into the classically processable 0 or 1 state. Therefore, it is necessary to measure qubits multiple times to retrieve a probability distribution describing the qubit's state. The number of repeated circuit executions is called *shots*, and a higher number of shots leads to a more precise probability distribution. As quantum circuits generally comprise multiple qubits, the measurement is described using a bit string, with each bit representing one qubit's measured state. The measured bit strings for each shot are counted and commonly referred to as *counts* or a *measurement result*. Inaccurate measurements of quantum states lead to a result that does not properly represent the prepared state. Hence, REM is performed to reduce the impact of measurement errors and to shift the result closer to the prepared state.

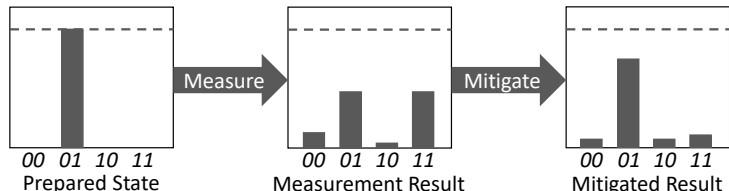

**Figure 1.** A high-level overview of the readout error mitigation process.

Typically, REM methods try to estimate measurement error rates and use these estimates to reduce the impact of errors on the measurement result. Since these error rates change over time, they are only valid for a limited time span [35–37]. To retrieve these so-called *calibration data*, a number of additional, method-specific circuits are executed. These mitigation circuits can either be independent of the *target circuit* that shall be mitigated, or they can be modified versions of that circuit. The former are reusable for different target circuits and are known as *calibration circuits* [34]. Commonly, their measurement results can be transformed into a reusable *mitigator* that is directly applied to a measurement result to mitigate it. Furthermore, the error rates differ for each qubit, making it important that the correct qubit mapping is used for error mitigation [15,17]. As all quantum circuits need to be transpiled for the quantum device's gate set and qubit connectivity map to execute them, it is of particular importance to use the qubit mapping of the transpiled circuit for

error mitigation. Otherwise, the mitigation methods might determine error rates for qubits that are not even measured when executing the quantum circuit.

### 2.2. Workflow-Based Modeling and Execution of Quantum Applications

Workflow technologies enable complex orchestrations of heterogeneous tasks, and have been proven to be applicable in different application domains, such as business process management or e-science [25,26]. As quantum applications typically comprise different classical and quantum tasks, workflows are also suitable for their orchestration [23,24,27]. These tasks can be quantum algorithm-independent, such as database access, or quantum algorithm-specific pre- and post-processing tasks, e.g., analyzing continued fractions for Shor's algorithm [38] or REM. Using workflows, the required tasks, their execution order, and the data flow between them are defined in so-called *workflow models*, which are automatically enacted using compatible workflow engines [26,39]. Both classical and quantum tasks, as well as the behavior in the case of certain events, e.g., timeouts or errors, can be modeled and automatically executed using workflows. Thus, quantum applications can benefit from their advantages, such as robustness, reliability, and scalability [23].

To facilitate modeling quantum tasks in workflows, Weder et al. [27] introduce the *Quantum Modeling Extension (QuantME)*, which can be applied to various imperative workflow languages, such as BPMN [28] or BPEL [40]. Thereby, new so-called *QuantME tasks* are introduced alongside typical configuration options for commonly occurring tasks in the quantum computing domain, such as quantum circuit loading, quantum circuit execution, and REM. Figure 2 shows a typical workflow model for executing a *Variational Quantum Algorithm (VQA)* [41]. VQAs alternate between executing a parameterized quantum circuit on a quantum device and optimizing the circuit parameters classically on the basis of the measurement results. The workflow model uses native BPMN modeling constructs, as well as the QuantME tasks. It comprises several tasks that must be executed to obtain the algorithm result. First, the parameterized quantum circuit solving the given problem is loaded by the *Load Circuit* task. The QuantME quantum circuit loading task is configured by either providing a URL to a file containing a circuit, or by directly inserting the code of a quantum circuit into the workflow model. Next, the *Execute Circuit* task executes the quantum circuit on the quantum device for the first time. It requires the specification of a provider and QPU name, and the number of shots used for execution. Once the circuit execution is completed, the *Perform REM* task is executed to apply REM to the retrieved measurement result. The definition of a REM method and a QPU are required by the QuantME REM task. Optionally, the maximum age for calibration data can be set. However, additional configuration properties, e.g., method-specific properties, are currently not supported. Afterwards, the *Evaluate Result* tas processes the mitigated measurement result by evaluating an objective function, which rates the quality of the retrieved measurement result [42]. If the evaluation shows that the quality of the result is not yet converging and the maximum number of optimization steps is not reached yet, the *Optimize Parameters* task is executed, which computes new parameters for the quantum circuit and invokes the *Execute Circuit* task once again. Otherwise, the quantum algorithm execution is completed, and the user can evaluate the result in the *Analyze Result* task.

However, the newly introduced QuantME tasks would reduce the portability of the workflow models, as the workflow engines must be extended to support their processing [29]. Therefore, all QuantME tasks are replaced by reusable workflow fragments [43], so-called *QuantME replacement fragments* [27], before executing the workflow. These QuantME replacement fragments implement the functionality of the respective QuantME tasks. Further, they only utilize native modeling constructs of the workflow language, and hence the portability of the workflow model is retained. Once modeled and transformed, the workflow model can be automatically executed by compatible *workflow engines* [26].

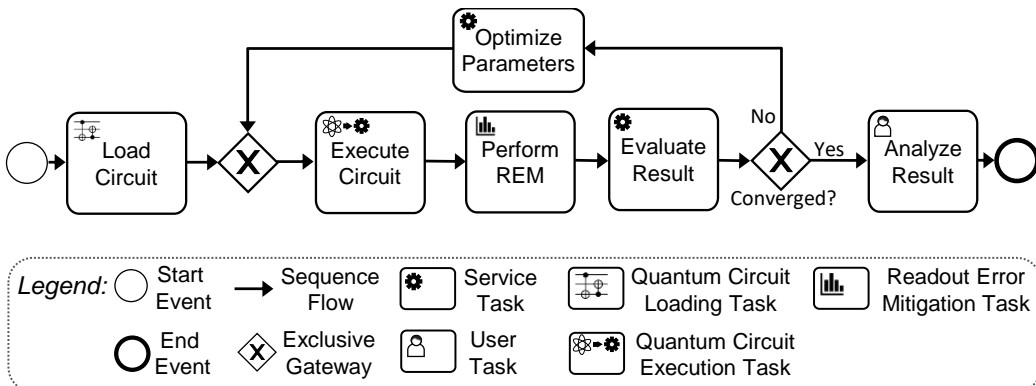

**Figure 2.** Overview of a typical QuantME BPMN model for a VQA.

## 3. Analysis of Configuration Options in Readout Error Mitigation Methods

In this section, we survey existing REM methods and analyze their configuration options to address Challenge 1 formulated in Section 1. We first present the survey design and then proceed with the analysis of existing REM methods and their configuration options.

### 3.1. Survey Design

This survey aims to collect existing state-of-the-art research literature that focuses on methods for mitigating the effects of measurement errors in quantum computations and that were published in 2021 or earlier. The main goal is to analyze and categorize the configuration options specific to each of the identified methods, such that the resulting options can be used for the automatic configuration of employed REM methods in quantum workflows. We design this survey following the existing guidelines for analyzing the academic literature [44–46]. Figure 3 shows an overview of the applied multiphase search and selection process. In the following, we elaborate on each phase, beginning with the initial search description.

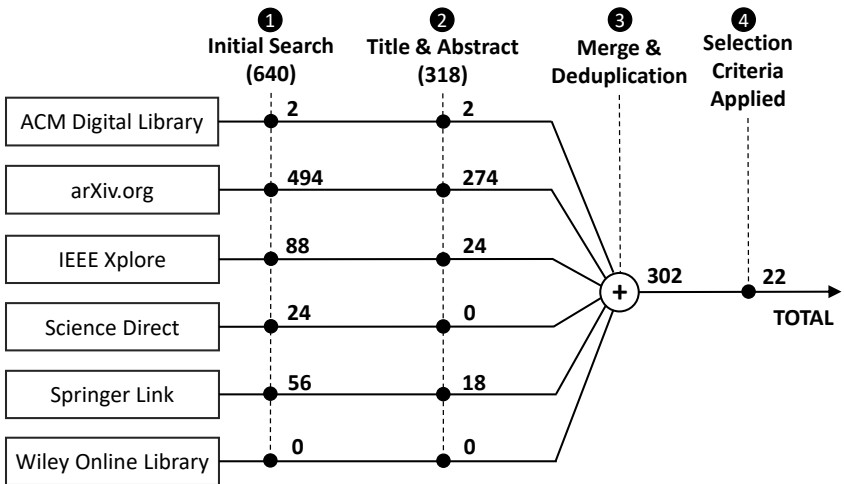

**Figure 3.** Overview of the search and filtering process (based on [47]).

1. **Data sources and initial search.** To identify existing REM methods, we scan the data sources recommended in well-established guidelines for a literature search [44–46]: We queried the following scientific electronic databases: (i) ACM Digital Library, (ii) arXiv.org, (iii) IEEE Xplore, (iv) Science Direct, (v) Springer Link, and (vi) Wiley Online Library. We decided to include arXiv.org, as many related publications focusing on the quantum domain are published as preprints. For the query, we defined a generic search string to cover a wide range of error mitigation methods used in the quantum domain, as shown in Listing 1.

Listing 1: Query string used for the initial search

*quantum* **AND** (*error* **OR** *noise*) **AND** (*mitigation* **OR** *unfolding*)

Unfolding is used to describe the mitigation of noise in high energy physics, and is also applicable in the quantum computing context [34]. To further reduce the number of false positives in the Springer Link search results, the *NEAR* operator was employed instead of the second *AND* operator: The *NEAR* operator ensures that the term on its left side is within ten words of the term on its right side, which helps to reduce irrelevant results when searching for combinations such as "error mitigation" or "noise mitigation". The initial search resulted in a total of 640 entries.

2. **Screening based on Titles and Abstracts.** In the second phase, we pruned the retrieved literature by scanning the title and abstract on their relevance to the topic of quantum error mitigation. In unclear cases, we also analyzed the paper's content in order to reduce the number of false negatives, using the adaptive reading depth [45]. After screening the initial search results, we identified 318 entries relevant to quantum error mitigation.

3. **Merge and De-duplication.** After identifying the initial set of relevant publications, we merged all entries from the different scientific electronic databases into a single data set. Then, we removed all duplicates based on the combination of title, authors, and venue. Thereby, we prioritized peer-reviewed papers published in journals or at conferences over pre-prints. In total, 302 entries remained after the merge and de-duplication process.

4. **Applying the Selection Criteria** In the final phase, we filtered the entries based on a set of selection criteria. For the analysis, we used the adaptive reading depth strategy [45], as it was sufficient to categorize the majority of the entries. In unclear cases, the relevance was discussed by all co-authors until a consensus was reached. We applied the following inclusion (✓) and exclusion (✗) criteria to the set of remaining publications:

   ✓ Publications that introduce new REM methods for gate-based quantum devices.

   ✓ Publications that are written in English.

   ✗ Publications that introduce methods that are based on hardware modifications.

   ✗ Publications that introduce methods focusing on error correction, mitigating gate errors, or on reducing the overall error, instead of specifically focusing on measurement errors.

   ✗ Publications that present use-case-specific methods, e.g., REM approaches in the context of quantum chemistry that are not generally applicable.

   ✗ Publications that use, benchmark, evaluate, review, or compare existing methods.

   ✗ Publications that are not available in the form of a full research paper, e.g., presentation, tutorial, abstract, or book.

   The resulting set of entries contains 22 publications. The REM methods introduced in these papers are briefly discussed in the following sections.

*3.2. Calibration Matrix-Based Mitigation Methods*

A large number of identified REM methods are related to the concept of *Calibration Matrices (CMs)*. A CM contains information about the measurement error rates for each of the basis states. As there are $2^n$ basis states for n qubits, the size of the matrix is $2^n \times 2^n$, with each cell containing the probability for a prepared basis state in column $a$, to be measured as the basis state in row $b$. Consequently, all values of a column sum up to 1, as each state is guaranteed to be measured as one of the basis states. Low measurement error rates are indicated by values close to 1 at the diagonal of the CM, while values significantly lower than 1 indicate high error rates. As the measurement result is a vector of counts, the composition of the noisy measurement can be described as follows: $CM * M_{True} = M_{Noisy}$,

where $M_{True}$ is the theoretical error-free measurement and $M_{Noisy}$ is the noisy measurement result obtained from executing the circuit on a noisy quantum device. Since we are looking for a way to transform the noisy measurement into the true measurement, we need to transpose the equation as follows: $M_{True} = CM^{-1} * M_{Noisy}$. Therefore, in principle, it is possible to retrieve the true probability distribution by accurately identifying the CM and inverting it [34,48]. The inverse of a CM is a typical example of a reusable mitigator. Since the number of matrix entries scales exponentially with the number of qubits, CM-based methods that are not specifically focusing to improve the scalability are not feasible for executions with a large number of qubits. In the following, we showcase different CM-based REM methods, whereas methods that exclusively focus on generating new CMs are presented in Section 3.3.

### 3.2.1. Calibration Matrix Subspace Methods

These methods focus on providing a good mitigation quality while using only the sub-matrix of the CM. The advantages over CM-based methods requiring a full CM are significantly lower classical resource requirements and improved scalability.

1.  **Perturbative REM for Near-Term Quantum Computers** [49]: This method offers resource-efficient REM via post-processing, by only employing a part of the full CM. Under the assumption that the probability of many simultaneous bit-flips during one measurement operation is low, it is sufficient to only use a small sub-matrix of the CM to reduce the device's error rate. Therefore, this method reduces the overhead for error mitigation; however, its application is limited to scenarios where measurement errors are not the main error source. No reference to an implementation is provided.
2.  **MThree** [17]: The **M**atrix-free **M**easurement **M**itigation (MThree) routine works on a reduced subspace of the CM. For large devices, the number of unique measured bit-strings is generally significantly smaller than the number of all possible bit-strings. Therefore, it makes sense to define the subspace used for mitigation by the measured noisy bit-strings, leading to a reduced dimensionality. The resulting equation system can then be solved much more efficiently than one for inverting the full CM. MThree uses its own custom calibration data generation procedure that for $n$ qubits requires $2n$ circuits. The method's implementation is publicly available on GitHub [50].

### 3.2.2. Optimization-Based Mitigation Methods

Many computational problems can be formulated as optimization problems, e.g., finding the maximum of a mathematical function. To solve such problems, several optimization procedures, e.g., COBYLA, have been developed that optimize the variables influencing the solution [51]. These optimization procedures minimize or maximize the value of an objective function describing the optimization problem, by systematically evaluating different parameter values. An example of choosing parameters are gradient descent algorithms [52]. For REM, optimization-based mitigation methods can be used to optimize an objective function that describes the reconstruction of the true, error-free measurement result.

1.  **Genetic-Based REM [20,53]:** Genetic-based REM methods build upon optimization-based evolutionary algorithms. These methods determine mitigator candidates by manipulating the set of potential solutions, the so-called chromosomes. Initially, a set of chromosomes is randomly created, and its fitness is evaluated, e.g., by means of the mean squared error. With every evolution of the algorithm, the fittest chromosome, i.e., the best solution, is carried over and mutated to generate similar and even fitter solutions. This process is repeated until a sufficiently good chromosome is found. No reference to an implementation is provided.

### 3.3. Calibration Matrix Generation Methods

In this section, we describe methods that can be used to generate a CM. In addition to using different calibration circuits, the discussed methods also differ in the number of circuits they execute and the number of error rates they use to build the CM. Generally, a

higher number of calibration circuits and retrieved error rates increases the accuracy of the CM. However, a high number of calibration circuits leads to a more time-consuming and expensive quantum circuit execution process, whereas a higher number of error rates increases the required storage size for the CM. The mitigation process itself is performed by applying one of the CM-based REM methods introduced in Section 3.2.

1.  **Full Calibration Matrix Generation [15,34]:** This method generates a CM in the most basic way and is commonly used as a benchmark for more advanced methods. First, $2^n$ shallow calibration circuits are generated, each of them preparing one of the basis states, immediately followed by a measurement operation on all qubits. Next, all circuits are executed, and the obtained probabilities for measuring each of the basis states are used as values for each field of the column representing the circuit's prepared basis state. Therefore, the CM has a size of $2^n \times 2^n$. The high number of calibration circuits makes this method infeasible for quantum devices with more than a few qubits. An exemplary implementation can be found in Qiskit Ignis [54].

2.  **Tensor Product Noise Model (TPNM) [15]:** The TPNM method focuses on mitigating single-qubit measurement errors, and assumes that the measurement errors of all qubits are independent. Therefore, it is sufficient to generate two calibration circuits to determine each qubit's error rate, $\epsilon_i$ and $\eta_i$, characterizing the measurement errors $|0\rangle \rightarrow |1\rangle$ and $|1\rangle \rightarrow |0\rangle$, respectively. With these error rates, the following tensor product-based CM of size $n \cdot 2 \times 2$ can be formed:

$$TPNM\ CM = \begin{bmatrix} 1 - \epsilon_1 & \eta_1 \\ \epsilon_1 & 1 - \eta_1 \end{bmatrix} \bigotimes ... \bigotimes \begin{bmatrix} 1 - \epsilon_n & \eta_n \\ \epsilon_n & 1 - \eta_n \end{bmatrix}$$

    Thus, it is possible to generate a mitigation model on the basis of $2n$ error rates, instead of the $2^n$ error rates that are required by the full CM generation procedure. This method is, e.g., implemented as a part of the Qiskit Ignis package [54].

3.  **Continuous Time Markov Processes (CTMP) [15]:** The CTMP method extends the TPNM method by including crosstalk into the error model. This is achieved by not only looking at each qubit's measurement error independently, but also including their respective 2-qubit measurement errors. Thus, the number of required calibration circuits increases from 2 to less than $2n$, which, however, is still exponentially less than the $2^n$ circuits required by the full CM generation procedure. The CM $M$ is in the form of a matrix exponential $M = e^G$, with $G$ being the sum of the $2n^2$ error rates. This method is, e.g., implemented as a part of the Qiskit Ignis package [54].

4.  **Diagonal Detector Overlapping Tomography (DDOT) [55]:** The DDOT method is used to generate a CM based on a generalization of the quantum overlapping tomography introduced by Cotler and Wilczek [56]. This procedure clusters strongly correlated qubits and then generates a noise matrix for each of these clusters. Further, clusters can have dependencies on other clusters, making them dependent on their pre-measurement state. Since these clusters can be examined in parallel and not all possible qubit correlations need to be examined, the number of circuits can be drastically reduced in comparison to the full CM generation procedure. In order to construct a $k$-local CM for N qubits, $O(k2^k log(N))$ circuits are required [55]. Thereby, $k$-local means that for every subset of qubits of size $k$, each of the basis states on that subset is prepared at least once [55]. An implementation of this method is provided by the authors on GitHub [57].

5.  **Conditionally Rigorous Mitigation of Measurement Errors [58]:** This method focuses on creating a CM that is free of any state-preparation errors. By applying single-qubit gate set tomography [59,60], an uncontaminated error analysis can be performed. However, this more precise mitigation scheme comes at the cost of $4^n$ measurements [58]. No reference to an implementation is provided.

6.  **QREM through Fuzzy C-Means Clustering [19]:** This method aims to generate a CM that is highly tolerant to the effect of stochasticity. First, a calibration circuit

for each basis state is executed $t$ times in order to create $2^n$ data sets, containing the probability distributions for each basis state, similar to the full CM generation method. In the second phase, the Fuzzy C-Means (FCM) algorithm [61] is applied for each of the $2^n$ data sets. The algorithm takes the data sets and a variable defining the number of clusters as inputs. A suitable value for the number of clusters can be determined, e.g., by applying the method introduced by Ross [62]. With these inputs, the REM method generates a fuzzy partition matrix by iteratively minimizing an objective function that optimizes the clustering for each data set. Finally, the CM is constructed by concatenating the most opportune probability vectors generated by the FCM algorithm. No reference to an implementation is provided.

7.  **Cumulant Calibration Matrix Construction [63]:** This method is based on a tensor product-based CM that can be generated using a linear number of circuits, similar to TPNM. However, this type of CM does not consider the correlations between the qubits. To detect and incorporate these correlations into the CM, the multi-qubit cumulant of the qubits is computed. For example, the 2-qubit cumulant is able to characterize the 2-qubit correlations, and can be generated on the basis of 1-and 2-qubit conditional probabilities, e.g., $\lambda_{a,b}(01|00) = p_{a,b}(01|00) - p_a(0|0) * p_b(1|0)$, where $\lambda_{a,b}$ is the 2-qubit cumulant for qubits $a$ and $b$ for the state transition $|01\rangle$ to $|00\rangle$, with $p$ denoting the corresponding conditional probabilities. Further, clusters of correlated qubits can be defined to reduce the number of qubit comparisons. Therefore, only the qubit correlations within a cluster are examined and applied to the CM. No reference to an implementation is provided.

8.  **Scalable T Matrix Estimation [64]:** To reduce the number of measurements circuits required for the full CM, this method estimates the CM on the basis of a tensor product and a neighborhood model. A qubit $i$'s neighborhood depends on the QPU's topology and includes the $k$ qubits with the shortest distance. To estimate the CM, each qubit and its neighbors are measured in the $|0\rangle$ and $|1\rangle$ state, resulting in $< 2n2^k$ measurements. Additionally, the pairs are measured, leading to a total of $< 2n^2 4^k$ measurements. With this data, the CM can be estimated by combining the single-qubit and pair measurement results. No reference to an implementation is provided.

*3.4. Circuit Modification-Based Mitigation Methods*

This group of REM methods focuses on reducing measurement errors by modifying the target circuit. Typically, a set of slightly modified versions of the target circuit is generated to retrieve information about the errors that occurred during its execution.

1.  **Active REM [65]:** This method mitigates measurement errors on a shot-by-shot basis, making it suitable for tasks not relying on the expectation value, such as Shor's algorithm [66]. Active REM is based on the concepts of error correction, but it only focuses on bit-flips in the computational basis occurring during measurement. Each qubit's state is encoded into multi-qubit states by entangling it with one or multiple ancilla qubits right before the measurement. Thus, it is possible to detect measurement errors and to correct them with a majority vote if two or more ancilla qubits are used. Further codes are introduced that perform a similar procedure for multiple qubits with fewer ancilla qubits [65,67]. No reference to an implementation is provided.

2.  **Hybrid Quantum-Classical Approach to REM [68]:** This method comprises two main steps. Firstly, the quantum circuit is adjusted by adding single-qubit gates performing a collective channel-twirl, before the circuit's measurement operations. When performing a channel-twirl, with every circuit execution, a randomly chosen gate is selected from a set of twirling unitaries, resulting in a conjugation of the noise (for more details, see [68,69]). Therefore, the channel-twirl depolarizes the qubits, making the measurement in the computational basis optimal. In the second step, classical post-processing is used to mitigate the target circuit's measurements based on the observed error characteristics and estimated device errors. No reference to an implementation is provided.

3.  **Crosstalk-Focused REM Protocol [70]:** This method mitigates individual measurement errors and crosstalk errors occurring during the measurement using a combination of circuit modification and classical post-processing. First, parameterized quantum gates are inserted into the quantum circuit right before the measurement operation. The parameters for these gates are determined by evaluating the measurement characteristics obtained via quantum detector tomography [71]. Moreover, these measurement characteristics are used for classical post-processing to further reduce the measurement errors. No reference to an implementation is provided.

Some of the identified REM methods rely on pre-measurement bit-flips to mitigate the impact of measurement errors. These methods exploit the phenomenon of lower measurement error rates for qubits in the $|0\rangle$ state, compared to the $|1\rangle$ state.

4.  **Static Invert-and-Measure (SIM) [21]:** The SIM REM method mitigates the impact of measurement errors by executing one or multiple versions of the target circuit, including bit-flips right before the measurement operation, in addition to the execution of the target circuit. Suggested bit-flip patterns are: a bit-flip for every qubit, or for every second qubit. Lastly, the measurements are classically analyzed and combined in order to filter out the measurement errors. May et al. [72] introduce similar approaches. No reference to an implementation is provided.

5.  **Adaptive Invert-and-Measure (AIM) [21]:** This method analyzes and learns the relative biases of different states, based on device-specific error rates, to place bit-flips at suitable qubits. First, the device characteristics are learned, e.g., by a CM. Then, several circuit executions are performed to estimate each qubit's bias and to determine the most fitting bit-flip positions. Finally, the circuit including the pre-measurement bit-flips is executed and evaluated. No reference to an implementation is provided.

6.  **Bit-flip Averaging (BFA) [73]:** BFA flips random qubits prior to the measurement with every shot. Immediately after the measurement, another bit-flip is applied classically to restore the original state. Generally, this removes all measurement biases from the measurement process. This bias-free measurement can also be used to simplify the CM generation process. No reference to an implementation is provided.

7.  **Model-Free Readout Error Mitigation for Quantum Expectation Values [74]:** This method focuses on mitigating correlated readout errors efficiently. First, general information about the error rates is collected by executing benchmarking circuits that contain random bit-flips prior to the state measurement. Second, circuit-specific data are collected by generating modified versions of the target circuit containing random bit-flips and executing them. Finally, readout errors are mitigated by evaluating the data retrieved for the target circuit and the benchmarking circuits. The information collected in step one can be reused for different target circuits. No reference to an implementation is provided.

*3.5. Expectation Value-Based Methods*

In this section, we present methods approximating a mitigated expectation value.

1.  **Measurement Error Mitigation via Truncated Neumann Series [75]:** This method bypasses the problems of inverting exponentially growing CMs by approximating the inverted CM with a truncated Neumann Series. This series is obtained by combining multiple sequentially measured noisy expectation values. Therefore, this method is only viable for algorithms that are based on the expectation value, such as VQE [76] or QAOA [77]. No reference to an implementation is provided.

2.  **Local, Spatially Uncorrelated Measurement Error Model (LSU) [78]:** This method computes the expectation value via a simple error model. First, for each qubit $i$, the probabilities of unexpectedly flipping from $|0\rangle$ to $|1\rangle$ and vice versa are determined,

as $p_i(1|0)$ and $p_i(0|1)$, respectively. Afterwards, these are used to correct the result's expectation value $E$ by applying the following formula:

$$E(counts) = \sum_{x \in counts} p(x) \prod_{i=1}^{len(x)} \frac{(-1)^{x_i} - p_i^-}{1 - p_i^+}$$

$p(x)$ is the probability of measuring the bit-string $x$, $x_i$ are the bit-string's individual bits, and $p_i^{\pm} = p_i(0|1) \pm p_i(1|0)$. Summing up the values for each bit-string returns the mitigated expectation value. No reference to an implementation is provided.

### 3.6. Machine Learning-Based Methods

Machine learning-based REM methods mitigate errors by learning a device's error behavior using typical machine learning techniques, such as neural networks.

1.  **Deep Neural Network Readout Error Mitigation [22]:** This method employs a deep neural network (DNN) for the characterization of a device's error rates. Deep learning is known for finding non-linear effects in a data set, making it a good candidate for mitigating non-linear measurement errors. To mitigate measurement errors, the DNN first needs to be constructed and trained. The input and output layers have $2^n$ nodes, whose values represent the measurement probabilities. Further, several fully connected hidden layers are added in between. These employ an activation function that is trained to be the inverse of the true error mapping. The training is performed by running various simple circuits with known optimal solutions. Once the DNN is set up, a mitigated measurement for circuits without a known optimal solution can be determined by the DNN. No reference to an implementation is provided.

### 3.7. Summary of Categorized Configuration Options

The survey of REM methods shows that the configuration options vary notably across different method categories. In the following, we discuss the identified configuration options and highlight other characteristics that need to be considered when performing REM.

-   **Configurations for Quantum Device Selection:** The same provider and QPU must be used to perform REM as are used to execute the target circuit. Therefore, REM-specific circuits, e.g., calibration circuits, must be generated in a language that is compatible with the chosen QPU. Further, some QPU providers already offer a set of publicly available calibration data that can be taken advantage of during the REM process [37]. For example, IBM provides data for each qubit's bit-flip probability during measurement. This data can be used by methods that focus on uncorrelated single-qubit measurement errors, such as the TPNM method.
-   **Configurations for Quantum Circuit Execution:** Due to the probabilistic nature of quantum computing, quantum circuits need to be run multiple times on the quantum device to obtain a reliable probability distribution. The number of shots used for execution depends on the number of measured qubits $n$, as the sampled state space scales with $2^n$ and consequently the number of samples must be increased. Thus, a suitable number of executions needs to be chosen to not retrieve an imprecise probability distribution or to waste quantum resources.

    Furthermore, since the set of measured qubits can differ between the target circuit's non-transpiled and transpiled versions, it is crucial for many REM methods to know the mapping of the logical qubits to the physical qubits. As each qubit's error rate differs, determining calibration data for the wrong set of qubits can lead to inaccurate mitigation results if the qubits' error rates differ significantly. Explicitly considering the transpiled version of the target circuit is particularly important for methods that determine error rates independently of the target circuit, e.g., CM-based methods or learning-based methods. Otherwise, it is unclear if the error rates were determined for qubits that were actually measured by the target circuit. Whereas, for example,

bit-flip-based methods, such as BFA and SIM, automatically operate on the correct qubit, as the bit-flips are part of the modified target circuit.

Finally, QPU providers typically require users to log in with their user account or to authenticate themselves with a token, to execute circuits on available quantum devices via the cloud. For example, to generate a CM for an IBM device, a valid token providing access to the selected device must be included in the circuit execution request.

- **Method-Specific Configurations:** Reusing existing calibration data and mitigators can significantly speed up the mitigation process and lower costs, as no new quantum circuits need to be executed. However, reusing existing data is only possible for REM methods that prepare a target circuit-independent error model of the device, e.g., CM-based methods, since the generated calibration data and mitigators can also be used for other target circuits run on the same device and with the same set of qubits. For this purpose, calibration data and mitigators must be stored and annotated with metadata, such as the device name, the set of qubits, and a timestamp. Future REM process instances can check for suitable calibration data and mitigators, instead of generating new data. In particular, the NISQ-typical VQAs benefit from reusing existing data, as they repeatedly run similar circuits on the same QPU within a short period of time. Due to continuously changing error rates, users must be careful not to use outdated mitigation data [36,37]. This can be supported by limiting the maximum age of existing calibration data. For example, when using QAOA for solving an optimization problem, the same parameterized quantum circuit is run multiple times in a loop. After each circuit execution, REM is performed. When using a reusable REM method, such as TPNM, and when there is no suitable mitigator available, it is sufficient to execute the calibration circuits to obtain a CM containing the calibration data and compute a mitigator, e.g., by inverting the CM only once in the first iteration of the QAOA algorithm. For each subsequent iteration, the mitigator can be reused, assuming that the same set of qubits is used, and the time in-between the iterations does not exceed the set maximum age for the mitigator. On the contrary, target circuit-dependent REM methods, such as Active REM [65], which mitigate errors based on the target circuit, in most cases do not produce reusable calibration data, making it necessary to execute additional REM circuits with every iteration of the algorithm. Therefore, configuration options focusing on the reusability and storage of calibration data and mitigators are not relevant for these methods.

  Furthermore, a variety of methods presented in the survey involve method-specific parameters. For example, when using the DNN REM method, the number of hidden layers and the activation function need to be set up. Another example is the scalable T matrix estimation method, which requires the configuration of the neighborhood range used for mitigation. Looking at optimization-based methods, such as the genetic-based REM method, the chosen optimization function, e.g., COBYLA or SPSA, can drastically impact the runtime and quality of the REM process. Further, different objective functions, rating the quality of the measured solution, can be selected. For example, these can emphasize or ignore parts of the solution to facilitate the optimization process for the optimization function [42].

- **Restrictions:** The REM process may have certain restrictions that are introduced by the process requirements or the available hardware. In addition to high quality, other aspects must also be taken into account, e.g., costs and processing time. Hence, the solution providing the highest quality is not always the one that is the most interesting. In specific use-cases, e.g., time-critical computations, such as traffic flow control [79], the maximum processing time is more important than precision, since an outdated solution is of no use. The choice of REM method can also influence the costs for quantum and classical resources, as new data might need to be generated and processed. Therefore, cost constraints must also be considered when setting up REM. REM can not only be limited by the available quantum hardware and business decisions, but also by the available classical resources that are required for post-processing.

For example, CMs grow large in size for quantum circuits with a high number of qubits as their size scales exponentially (see Section 3.2). Similar, processing times for operations such as matrix inversion can also be very high. Hence, keeping hardware constraints in mind when performing REM can prevent long processing times, out of memory, and insufficient storage exceptions during the mitigation process.

## 4. Automating Readout Error Mitigation in Quantum Workflows

The survey of REM methods and their configuration options presented in Section 3 highlights a variety of general and method-specific options that must be considered when utilizing REM. In this section, we show how this configuration process can be automated to avoid manual, more error-prone configuration tasks and facilitate the reuse of resulting configurations for application developers, hence, addressing Challenge 2 formulated in Section 1. The focus is to provide these functionalities to workflow-based quantum applications, by building our approach on top of QuantME [27]. First, we discuss the process of automated workflow model refinement, which enables finding executable REM workflow fragments based on provided method configurations and reusing them as parts of general workflow-based executions of quantum applications. Next, we elaborate how the identified configuration options are mapped to different phases of the presented automated REM process. As examples, we introduce and discuss two REM workflow fragments that enable automatically executing methods from two REM categories discussed in Section 3, namely (i) matrix-based and (ii) circuit modification-based REM methods. Finally, we present a system architecture enabling our approach and elaborate on its prototypical implementation.

### 4.1. Overview of the Approach

From a modeling perspective, the REM configuration involves multiple manual tasks: depending on the chosen REM method and its respective implementation, the mitigation control flow and configuration options vary significantly, as shown in Section 3. Therefore, to reduce time-consuming and error-prone manual configuration tasks, we automate the REM configuration, as shown in Figure 4. The configuration process involves automated steps for both workflow modeling and execution. In the first step of the process, workflow modelers select a REM method. To enable transitioning to executable REM workflow fragments at design time, we extend the QuantME REM task to enable the specification of all configuration details for REM. Thus, the modelers provide the method- and process-specific information by configuring the QuantME REM task's options (Step 2). Table 1 shows the configuration options described in Section 3.7 and provides an overview of the corresponding data types and exemplary values. To facilitate the configuration of REM options, a list of suitable options for a chosen REM method can be provided, e.g., a selection of available QPU providers or usable mitigation methods. As a result, after Step 2, the QuantME REM tasks contain general preferences regarding the selected REM method and its configuration. Generally, options can be set during design time and execution time; e.g., the mitigation method and method-specific configurations are typically specified during the design time, whereas the target circuit's measurement result or the time of execution is not known during design time and has to be provided at execution time.

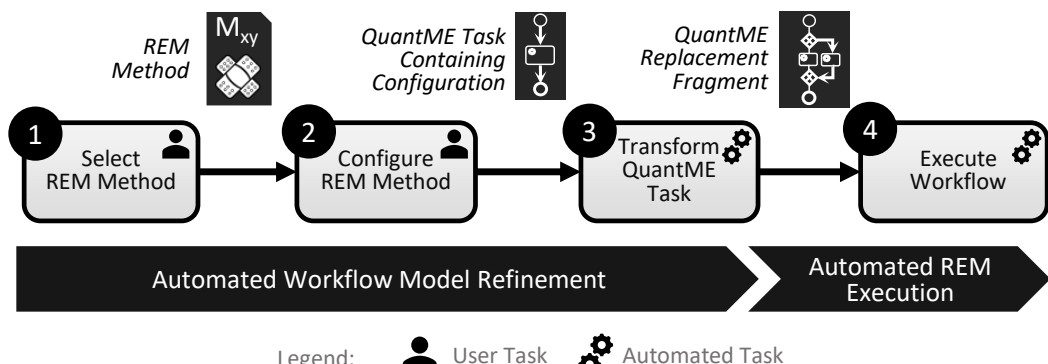

**Figure 4.** Overview of the REM process from a modeler's perspective.

**Table 1.** Overview of supported configuration options.

| Configuration Option | Datatype | Example |
| --- | --- | --- |
| Provider name | String | "IBM" |
| QPU name | String | "ibm_auckland" |
| QPU credentials | Dict of QPU-specific credentials | {"token": "1m3isj902sb74cdfak3"} |
| Qubit mapping | List of measured qubits in order | [0,1,5,19,7,6] |
| Shots | Integer | 1000 |
| Counts * | Dict of results to mitigate | {"00": 4, "01": 33, "10": 2, "11":61} |
| Mitigation method | String | "Inversion" |
| Calibration method | String | "TPNM" |
| Method-specific values | Dict of method-specific fields | {"#DNNLayers": 7, "activFunction": ...} |
| Max age (in minutes) | Integer | 720 |
| Max REM costs (in $) | Integer | 25 |
| Max CM size (in MB) | Integer | 500 |
| Time of execution | Datetime | 2022-04-22 11:23:57 |

* Synonymous to the term "measurement result", e.g., Qiskit "counts" or Braket "measurement_counts" (see Section 2.1 for more details).

Since the QuantME REM task is not directly executable, in Step 3, the QuantME REM task is transformed by means of QuantME replacement fragments [27]. In this transformation step, the configuration enabled by the QuantME REM task, which abstracts away all complex configuration and implementation details on the modeling level, is linked with the actual programs handling the execution of the configured REM method. To find a suitable replacement fragment, a repository containing QuantME replacement fragments added by domain experts is searched. For each of the fragments, it is checked whether the selected method and configuration choices are supported. Once a suitable replacement fragment is found, the QuantME REM task is replaced by it. Different REM methods are typically implemented by different QuantME replacement fragments, as their execution steps and implementation logic differ. In addition to describing the REM steps and their order, the replacement fragment must link the modeled functionality with implementations providing the functionality. Hence, the availability of these implementations must be ensured, e.g., by deploying them with the workflow. The implementations themselves can range from small scripts implementing a single REM method over services implementing a variety of REM methods. As generally any REM implementation can be integrated via a service into a replacement fragment, versatility and extensibility are ensured.

In the automated REM execution phase (Step 4), all required services are deployed [80,81] and the retrieved replacement fragment is executed as part of the overall quantum application workflow using a workflow engine. Furthermore, the configured QuantME replacement fragment can be saved and reused by future applications.

Figure 5 shows the steps of a CM-based REM method, including tasks for the reuse of calibration data and mitigators. First, it is checked if there is a suitable mitigator, i.e.,

a mitigator that is computed for the selected device is not outdated and fits the set of measured qubits. If a suitable mitigator is found, it is retrieved from the data storage and subsequently employed to mitigate errors in the noisy measurement. In case no suitable mitigator is found, the next step is to check whether suitable calibration data that can be used to compute a mitigator without executing additional circuits on the quantum device is available. If there is recent calibration data for the selected device and measurement qubits, it is retrieved from the data storage and subsequently used to compute a mitigator, which is saved in the data storage for future use. Afterwards, the mitigator is employed to mitigate errors in the noisy measurement. In case there is no suitable calibration data available, new information about the QPU's error rates needs to be retrieved. Hence, a set of method-specific calibration circuits is generated and subsequently executed on the QPU. Based on the measurement results of these calibration circuit executions, a mitigator is generated, which is eventually used to mitigate the errors in the noisy measurement results. To enable reuse of the generated data, they are saved in a data storage. It is important that the data are annotated with all necessary metadata, such as the used QPU and set of qubits, and the date and time of the calibration.

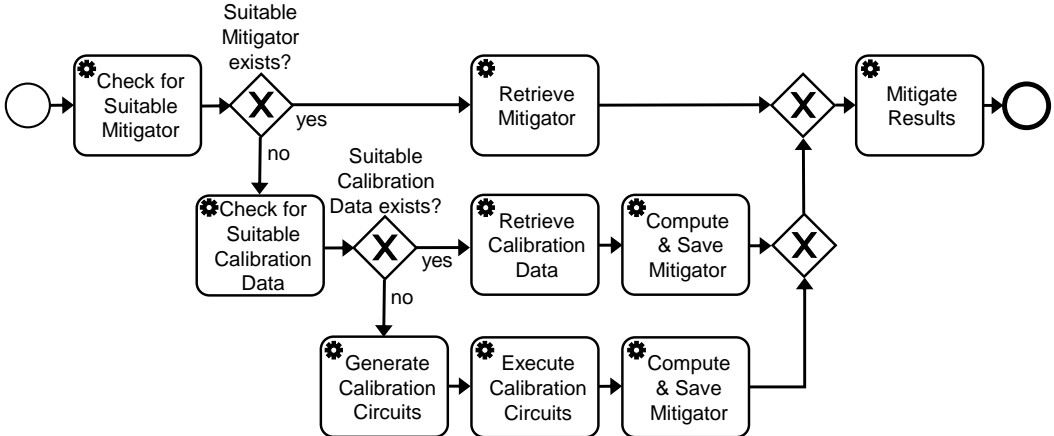

**Figure 5.** Exemplary REM workflow fragment for CM-based methods.

Figure 6 shows the steps of circuit modification-based REM method using bit-flips. Since the mitigation process is optimized for a specific circuit and there is no assessment or use of general error rates, results from previous executions cannot be used. The workflow starts by computing suitable positions for the bit-flips. Next, new circuits containing premeasurement bit-flips at the determined qubits are generated by modifying the target circuit. Afterwards, the circuits are executed on the quantum device, and the bit-flips are classically reversed to restore the original result. Finally, the observed difference in error rates when measuring $|0\rangle$ and $|1\rangle$ can be used to mitigate the result.

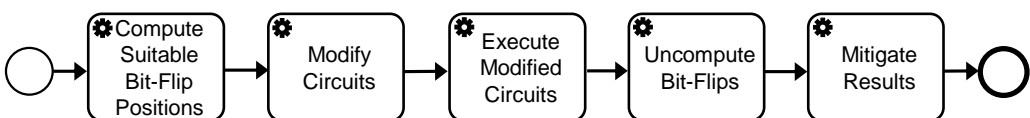

**Figure 6.** Exemplary REM workflow fragment for a bit-flip-based circuit modification method.

### 4.2. System Architecture

Our overall system architecture for automating configurable REM is shown in Figure 7. It comprises a graphical workflow modeler, the extended *QuantME Transformation Framework* [82], and a workflow engine for workflow integration and execution. Further, a provenance system is integrated to collect data for provenance, and multiple quantum execution services are used for executing quantum circuits on different QPUs [37]. Additionally, the *REM Service* is implementing the functionalities required by the REM workflow fragments

presented in the previous section. New components are highlighted in black, extended components in dark gray, and unchanged components are highlighted in light gray.

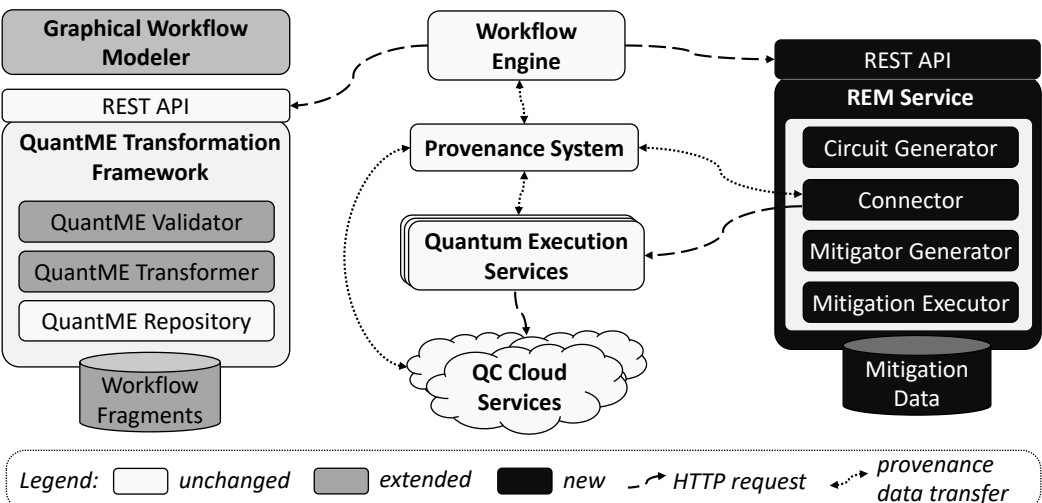

**Figure 7.** System architecture for automating REM configuration in quantum workflows.

The graphical workflow modeler enables the modeling of BPMN workflows containing QuantME tasks by means of a graphical user interface. Thereby, the QuantME REM task is extended to support the identified configuration options. The *QuantME Validator* ensures the validity of the QuantME tasks by highlighting errors during the modeling process in the graphical user interface. To guarantee the validity of modeled REM tasks, the QuantME Validator is extended by a new set of validation criteria, providing rules for the newly added REM configuration options. The *QuantME Transformer* transforms a QuantME workflow model into a native BPMN workflow model that can be executed by a BPMN workflow engine. The QuantME Repository is responsible for the management of QuantME-related data, such as QuantME replacement fragments.

The REM Service provides a REST API that can be used to trigger a REM process. The *Circuit Generator* generates the required quantum circuits for the REM method. These are executed on the selected quantum device via the *Connector* that triggers the circuit execution via an API call to the corresponding quantum execution service. These SDK-specific quantum execution services transpile the circuits and subsequently execute them on quantum devices provided via the cloud, e.g., IBMQ or Rigetti Quantum Cloud Services. An in-depth description of the execution services is provided by Salm et al. [83]. For target circuit-independent methods, the *Mitigator Generator* computes a mitigator on the basis of the calibration circuits' measurement results. Further, the mitigator is annotated with metadata and saved in the *Mitigation Data* repository for later reuse. The metadata can be used to identify suitable, already existing mitigators for subsequent REM requests instead of computing new ones. The *Mitigation Executor* mitigates the target circuit's measurement result by employing a mitigator or the REM method's circuits' measurement results. All the REM service's components are modularized; therefore, additional mitigation methods and execution services can be integrated easily.

A provenance system is used to continuously collect data from the workflow engine, the REM Service, the quantum execution services, and the quantum cloud services [37]. The collected data are saved in the long-term to enable the reproducibility and analysis of quantum applications. Furthermore, the data can be used to provide additional information supporting the REM process, e.g., when the last calibration of the QPU happened.

### 4.3. Prototypical Implementation

For the graphical modeling of quantum workflows in BPMN, the *QuantME Transformation Framework* [82] extends the *Camunda Modeler* [84], a JavaScript-based open-source

workflow modeler. It is provided as a standalone application and enables the modeling of QuantME tasks and their transformation using QuantME replacement models. QuantME tasks and replacement fragments are defined in BPMN XML syntax, and can be easily stored in a GitHub repository. To collect provenance data, we employ QProv [37], an open source, Java-based provenance system that is using the provenance standard PROV [85] to enable provenance for quantum applications.

Our implementation of the REM Service [86] supports all steps of the REM process, from the generation of REM-related circuits to the mitigation of the results. However, the individual REM steps could also be implemented differently, e.g., as separate services. Our REM Service is implemented prototypically in Python, using the Flask web framework to provide access to the service functionalities via an HTTP REST API. The API consists of endpoints for the generation of calibration data, the generation of mitigators, and for performing REM. The former two can be used to retrieve error rates and generate mitigators on a regular basis. Thereby, the evolution of hardware performance can be monitored, and suitable mitigators can be generated prior to or in parallel to the execution of the quantum application, speeding up the overall process. In addition to mitigating individual measurement results, the REM service is also capable of accepting bulks of measurement results. Thereby, the overhead that would be caused by a high number of individual requests can be reduced drastically. To guarantee quantum hardware independence, a common format is used for the measurement results. Thereby, the provider-specific result formats do not need to be taken care of for the implementation of the REM methods. For storing mitigators, we use MinIO [87], a distributed high-performance cloud object storage.

## 5. Case Study

To demonstrate the practical feasibility of our approach, we integrate it into an existing quantum workflow from the quantum humanities domain [88,89], which aims to detect patterns in costume data from movies, and enables the mapping of new costume data to these patterns. Thereby, clustering is used to partition the data into clusters; subsequently, a classification algorithm is executed to map new data to these clusters. Both the clustering and classification are implemented using quantum algorithms. To improve the result quality of these algorithms, we perform error mitigation after every circuit execution by extending the workflow model with our REM approach. The corresponding workflow is shown in Figure 8. The workflow model, implementation, and detailed instructions for setting up our case study are available on GitHub [90].

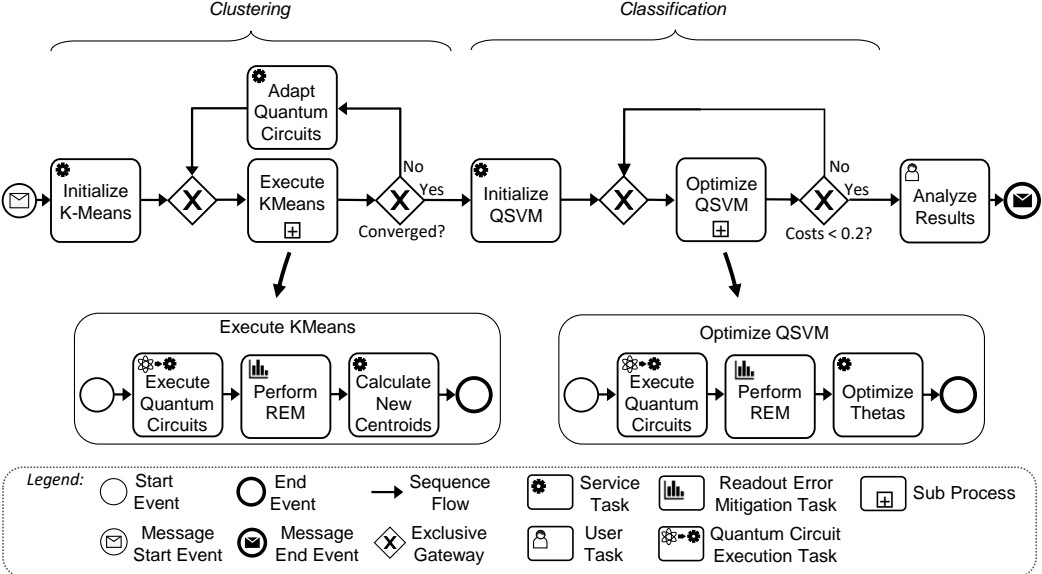

**Figure 8.** Exemplary workflow model from the quantum humanities domain.

For the integration of quantum tasks into the workflow, we use the BPMN extension Quantum4BPMN [27], which applies QuantME to BPMN. The workflow instantiation can be triggered by a message, e.g., via a simple HTTP request containing all required data. In Figure 8, the task of pre-processing the categorical costume data by transforming it into numerical data and reducing its dimension is not shown for brevity, and the workflow expects a URL to the prepared data as input instead (see [88] for details). Once instantiated, the workflow starts off by running the *quantum k-means-algorithm* [91], which consists of multiple tasks. First, the data is loaded from the URL, initial centroids are determined, and the initial set of circuits is created based on the centroids. Next, the clusters are optimized in a hybrid quantum-classical loop. Thereby, the "Execute KMeans" sub-process comprises the circuit execution, followed by REM, which improves the quality of the retrieved measurement results, and the computation of new centroids on the basis of the mitigated measurement results. The circuits are modified, and the loop is rerun until the difference between the old and new centroids is not significant anymore, i.e., smaller than a given threshold. Once the centroids converge, the final clustering is used as an input for the second hybrid quantum-classical loop, which performs classification using a variational *Quantum Support Vector Machine (QSVM)* [92]. Quantum circuits and initial parameters are generated, which are then once more optimized within the loop, following a similar approach as in the workflow's first optimization loop. Once the optimization process is finished, the result is returned via a message end event and can be analyzed by users.

REM is performed via the extended QuantME REM task described in the previous section. Figure 9 shows different steps of our case study execution. First, as depicted in Figure 9a, we selected the TPNM method for CM generation and matrix inversion to generate the mitigator. Next, in the REM configuration step, shown in Figure 9b, we chose IBM's *ibmq_lima* QPU and set the number of shots to 2000, the maximum matrix size to 512 MB, and the maximum age to 600 min. Information about the measurement results and the qubits used for measurement is provided to the REM task during workflow runtime. To utilize our prototypical implementation of the REM service via the QuantME task, we provided a suitable replacement fragment, managing the communication between the workflow engine and our REM service. It is integrated into the workflow in the transformation phase, shown in Figure 9c. Thereby, a replacement fragment repository provided via GitHub is checked for available replacement fragments, and a suitable one is chosen. This replacement fragment is executed during every iteration of the clustering and classification loop. It utilizes the REM service to retrieve a mitigator for the measurement qubits of the selected QPU. In the first iteration of the clustering loop, no suitable mitigator was available via the REM service's database yet. Therefore, two calibration circuits were executed whose measurement results were subsequently used to generate a mitigator that could be used to mitigate the target circuit's results. As the circuit in future iterations remained similar, the generated mitigator was reused and the mitigation overhead was minimal in comparison to the overall execution time. Once the workflow is transformed, it can be deployed and instantiated. During execution, the user can monitor the current state of the workflow instance. Figure 10 gives an overview of the *Camunda Cockpit*, the graphical user interface of the Camunda engine to monitor and analyze workflow executions. Thereby, the token flow within the workflow instance is visualized, i.e., the currently active task in the workflow is shown. Furthermore, it also displays the current values of the variables, e.g., comprising the counts of the quantum circuit execution before and after the mitigation or the number of the current iteration. When the workflow terminates, all collected data are moved to the *audit trail* and can be analyzed from there [26]. This enables, e.g., a comparison of the execution of quantum workflows using different REM methods and configurations, which can be the basis for adjusting their configurations accordingly for future executions.

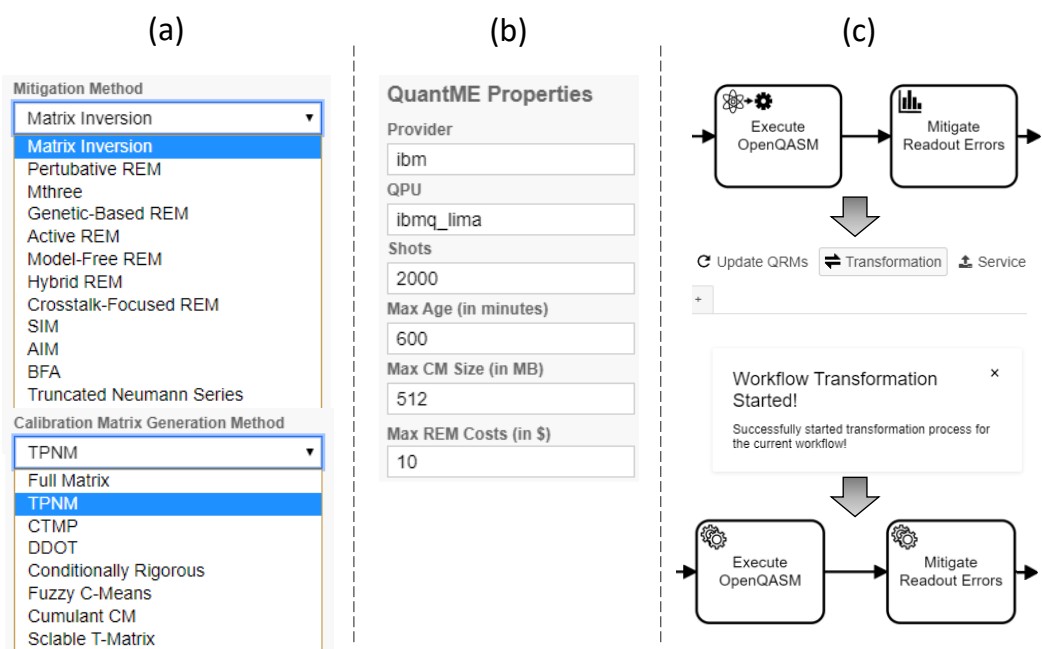

**Figure 9.** REM method selection (**a**), configuration (**b**), and workflow transformation (**c**) for our case study.

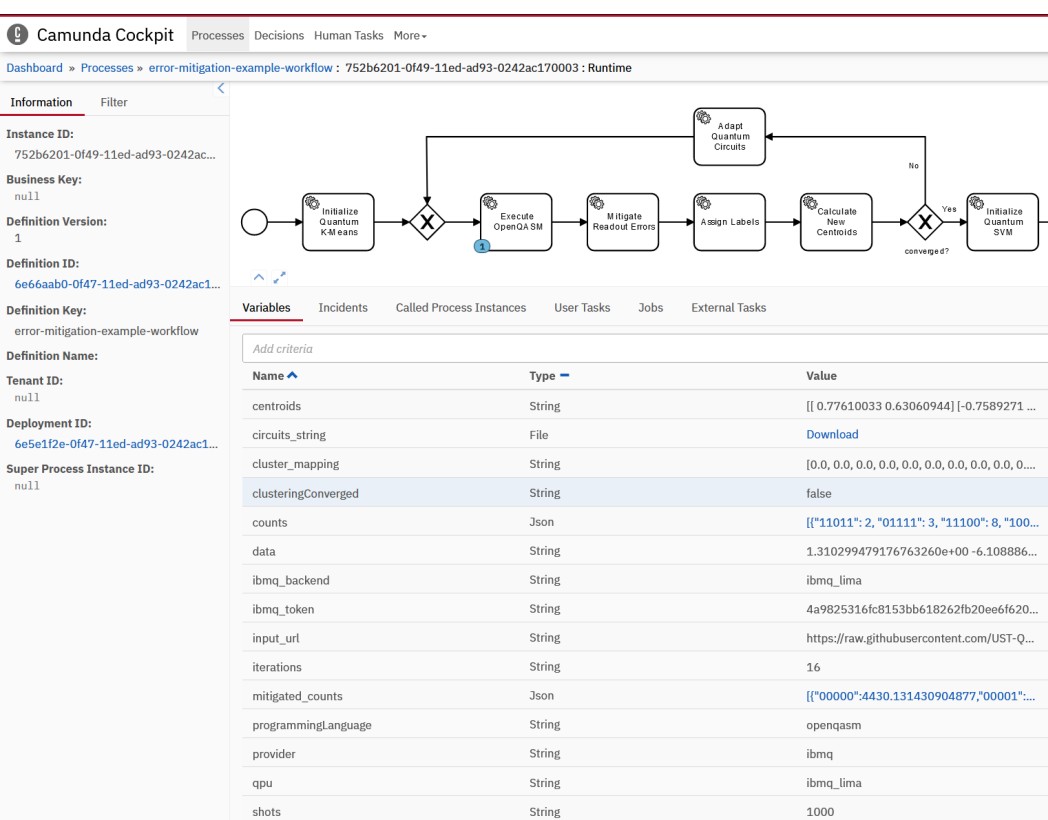

**Figure 10.** Overview of the case study's workflow execution in the Camunda user interface.

## 6. Discussion

In this section, we first discuss potential threats to the validity of the conducted survey and what measures were taken to mitigate these threats. Further, we evaluate our approach critically, focusing on the current state and developments in the quantum computing area.

### 6.1. Threats to the Survey's Validity

Conducting a survey is a challenging task, as the goal is to accurately capture all work fulfilling certain criteria in a huge pool of data. In the following, we elaborate on the key aspects threatening the validity of our survey.

**Selection bias.** One of the main threats to the survey of REM methods is that the primary set of publications is not representative enough and existing methods are not covered by the survey. To provide a broad set of initial data, we queried six well-established electronic research databases with a query consisting of high-level keywords. As a consequence, our initial data set was much larger than the final data set. To filter out the large number of false positives, we first scanned the papers by their title and abstract, and afterwards applied a set of selection criteria. To reduce the number of false negatives, we used the adaptive reading depth strategy [46] in unclear cases, and resolved conflicting opinions by discussing them internally. Further, we performed the survey with a team of six researchers, reducing the risk of an individual researcher wrongfully judging a paper.

**Reproducibility.** To enable other researchers to reproduce and verify our results, we described all steps performed during our survey in detail in Section 5.

**External validity.** In our study, we focused on peer-reviewed works from academia and pre-prints available via arXiv.org. Gray literature was not in the scope of this work. Therefore, we did not cover other knowledge sources, such as blog posts, code documentation, and other industrial efforts that are not published as academic research.

### 6.2. Extensibility and Limitations of the Automated REM Configuration in Quantum Workflows

The rapid developments in quantum computing require current quantum applications to be adaptable [93]. For example, new error mitigation methods are published on a regular basis, making the extensibility of our approach for new methods crucial. Further, the lack of well-defined standards leads to a high heterogeneity of quantum hardware and software development tools, which makes it non-trivial to provide a system with universal support. Additionally, there exist various workflow languages and engines that developers might want to use for realizing our approach. In the following, we discuss the extensibility and portability of our approach and system architecture for the aforementioned aspects.

1. **Adding new REM methods**: New REM methods can be added to our approach by integrating their configuration options into the QuantME REM task and ensuring that a suitable QuantME replacement fragment implementing the method is available in the replacement fragment repository. In the case of our prototypical implementation, we ensured that the QuantME tasks can easily be extended with new methods and configuration options, and that new REM methods can be included in the REM service as plugins implementing well-defined abstract functions.

2. **Adding new quantum providers**: New quantum providers can be integrated by providing an additional quantum execution service that is capable of executing a set of circuits and returning a list of results. This execution service is connected to the REM service via the execution connector component. Thereby, it is important that the language used by the new provider is already supported, otherwise further changes, which are discussed next, are necessary to enable the REM for this new provider.

3. **Adding new quantum programming languages**: Currently, there is no well-defined standard for the modeling and execution of quantum circuits. As a consequence, many quantum providers have developed their own languages, which are frequently incompatible with each other. Therefore, supporting all existing languages is a difficult task. In our prototype, we provided the support for Python with the Qiskit SDK and OpenQASM, which are employed by the majority of quantum providers and can be translated into many other languages [94]. Therefore, publicly available translators can be used to enable an easy integration of an additional language into the REM service. However, this comes at the cost of an increased amount of circuit translations. To natively support a new language in the REM service, the circuit implementations for all supported REM methods need to be implemented as plugins.

4. **Using a different workflow language or engine**: In past years, a wide variety of workflow languages and engines have been developed [95]. Although our prototype is implemented using BPMN and the Camunda engine, our approach is not limited to a specific workflow language or engine. For example, it is also compatible with BPEL or any other workflow language that (i) supports the definition of *tasks*, (ii) enables the configuration of tasks by *attributes*, (iii) allows the specification of a *control flow* (iv) and *data flow*, (v) and supports alternative control flows for error handling [27].

Furthermore, hybrid runtimes, e.g., Qiskit Runtime [96] and Amazon Braket Hybrid Jobs [97], enabling the execution of classical tasks close to the quantum device have emerged. Thereby, the latency for switching between the self-orchestrated classical tasks and the quantum circuit executions is reduced. The providers of these runtimes have announced the integration of automatic error handling into their hybrid runtimes, facilitating the application of error mitigation [98]. However, solely relying on the built-in solutions of a cloud provider limits developers' options to try other methods and leads to a vendor lock-in. Further, typical workflow advantages such as robustness, scalability, and monitoring capabilities are lost when using a hybrid runtime environment.

## 7. Related Work

The survey of REM methods provided in this work gives an overview of current REM methods and briefly describes how they work. In addition to this survey, there are other works showcasing and comparing REM methods [34,99]. However, they focus on a small number of methods and do not give a broad overview of available REM methods.

In previous work, we introduced quantum error handling patterns, showcasing different error handling approaches in an abstract manner [8]. Thereby, REM is introduced as a pattern for mitigating measurement errors in the NISQ era. The pattern highlights the context and forces of REM, and explains how REM generally works and how it can be integrated into a quantum application. Since patterns focus on presenting solutions in an abstract manner, the REM pattern only includes two concrete examples of REM methods. Thereby, the survey in this work can serve as an addition to the REM pattern, providing more methods and an approach for integrating REM into a real-world application.

Devitt et al. [100], Endo et al. [101], and Matsumoto and Hagiwara [102] survey different quantum error handling methods. However, they focus on quantum error correction and quantum gate error mitigation, and only briefly describe REM.

Various SDKs, such as Qiskit [103] and PennyLane [104] offer quantum error mitigation capabilities. Thereby, they implement individual methods, e.g., TPNM [15], or employ error mitigation frameworks, such as Mitiq [105]. Typically, these implementations are designed to be directly integrated into the code of the quantum application, instead of being provided as a service. Additionally, they focus on mitigating individual circuit executions and are not optimized for the execution of a high number of circuits. Furthermore, some method implementations expect SDK-specific objects and therefore are not platform-independent.

McCaskey et al. [106] present the hybrid programming model *eXtreme-scale ACCelerator (XACC)*. XACC enables the compilation and execution of quantum algorithms independent of their format and language. Further, XACC comprises a REM module, which can directly include REM into the execution process. However, this drastically limits the user's configuration and monitoring capabilities to the ones provided by XACC.

In 2020, Zapata Computing released *Orquestra*, a software platform for building and deploying quantum-ready applications [107]. These applications are built using so-called *quantum-enabled workflows*, using a YAML-based workflow language. To build a workflow, users have to define the required imports, e.g., a file in a Git repository containing application logic, and all steps of the workflow in the YAML file describing the workflow. Further, all information regarding the resources used for the orchestration of the tasks and the initial values of the parameters used for the execution of the tasks are also included in the same file. Moreover, Orquestra currently does not provide a tool enabling the graphical modeling of workflows to facilitate the modeling process, and does not provide a broad

set of common quantum task types comprising typical configuration options. Rather, users have to start building their workflow from scratch by means of writing a YAML file. In contrast, our approach uses standardized workflow models that can be executed on full-fledged workflow engines, providing advantages, such as robustness and scalability.

Another framework for executing quantum workflows is *Covalent* [108]. It is a Pythonic workflow tool specialized for executing tasks on HPC and quantum hardware. The workflow is generated by annotating the code with Covalent-specific decorators. In contrast to our approach, it is not based on a standardized workflow language and does not provide workflow-typical features such as transactions, error-compensation, and user tasks.

## 8. Conclusions and Future Work

A variety of error mitigation methods have been developed to improve the performance of today's noisy quantum devices. To facilitate the integration of REM in quantum applications, we first analyzed the literature for existing methods. We categorized the found methods and summarized their basic functionality to ease their understanding. Further, we evaluated the methods' configuration options to identify common and method-specific options that need to be considered when integrating REM into a quantum application. As quantum applications typically contain many quantum and classical software components, implementing, configuring, deploying, and orchestrating all components manually is error-prone and time-consuming. Thus, workflow technologies have been proposed as a means for orchestrating quantum applications. To automate the REM process in a configurable manner, we introduced an approach integrating service-based, configurable, and extensible REM into quantum workflows. To validate our approach, we provide a prototypical implementation and employ it in a case study from the quantum humanities domain.

In future work, we plan to further extend our prototype by providing accurate cost estimations for performing REM for different quantum providers. Furthermore, we will integrate REM in the NISQ Analyzer [83] to facilitate the hardware selection process. Moreover, we plan to compare the presented REM methods using a model-driven benchmarking approach. As hybrid runtimes, such as Qiskit Runtime [96] or Amazon Braket Hybrid Jobs [97], are becoming more popular, we plan to investigate how our workflow-based REM approach can be combined with current hybrid runtime environments. Finally, we plan to investigate whether our approach is also applicable to other types of error mitigation, such as gate error mitigation, and we extend our prototype to support these methods too.

**Author Contributions:** Conceptualization, M.B., J.B., F.L., F.T., B.W., and V.Y.; methodology, M.B., J.B., F.L., F.T., B.W., and V.Y.; software, M.B., F.T., and B.W.; validation, M.B., F.T., B.W., and V.Y.; investigation, M.B., F.T., B.W., and V.Y.; resources, M.B.; data curation, M.B.; writing—original draft preparation, M.B., F.T., B.W., and V.Y.; writing—review and editing, J.B. and F.L.; visualization, M.B., F.T., B.W., and V.Y.; supervision, J.B. and F.L.; project administration, J.B. and F.L.; funding acquisition, J.B. and F.L. All authors have read and agreed to the published version of the manuscript.

**Funding:** This work was partially funded by the BMWK projects *PlanQK* (01MK20005N), *EniQmA* (01MQ22007B), and *SeQuenC* (01MQ22009B), and by the project *SEQUOIA* funded by the Baden-Wuerttemberg Ministry of Economic Affairs, Labour and Tourism.

**Data Availability Statement:** The prototypical implementations and case study presented in this work are available on GitHub [82,86,90].

**Conflicts of Interest:** The authors declare no conflicts of interest.

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
