# Peer review of "Configurable Readout Error Mitigation in Quantum Workflows"

_electronics, doi:10.3390/electronics11192983_

Round 1

Reviewer 1 Report

1.The number of keywords in the abstract section is excessive, and it is recommended to control between 3 and 5.

2.Numbered descriptions are recommended when detailing the content of each phase in section 3.1.

3.This paper is lack of comparison with other measurement methods to illustrate the advantages of this paper.

4.The indentation of the paragraphs in this article is disordered, it is suggested to optimize a paragraph format.

5. It is better to add a text description about the picture.

Author Response

Thanks a lot for your valuable feedback.

We attached a file containing a detailed description of how we addressed each of the comments.

Reviewer 2 Report

In this draft, authors present an approach to reduce measurement errors (RME) in quantum devices. They firstly review existing methods and evaluate their general feasibility. The authors then propose their approach to abstractly model quantum workflows comprising configurable readout error mitigation tasks. These workflows can  be automatically refined into executable workflow models. To validate the feasibility, the authors provide a prototypical implementation and demonstrate it in a case study from the quantum humanities domain. The draft is organized well and it should be interesting to quantum information community. I would like to suggest the publications. The followings are my concerns.

1 I feel the abstract needs to be more concise.

2 There are some typos. For example, at line 491, there is a “f”. And at line 529, I think “t” should be capitalized.

Author Response

(The authors gave the same response as above.)

Reviewer 3 Report

In this work, Beisel et al. propose an approach to automate the REM process to reduce manual component error and time required for quantum computation. A summary and analysis of the literature on the existing methods are presented. Moreover, the proposed method is validated by implementing it in a case study from the quantum humanities domain, which involves clustering and classifying data. This is a proof-of-principal detailed study and is suitable for this journal. The authors have done a great job summarizing the existing literature and pointing out the validity and limitations of the current and proposed automated REM configuration methods. I could not find much to criticize except that the text can be more concise in a few places, especially in the early part of the manuscript. Overall, the writing is clear, and the observations are very well described, hence I recommend this work for publication.

Author Response

(The authors gave the same response as above.)
